# Impacts of speed and spacing on resistance in ship formations

Linying Chen
State Key Laboratory of Maritime
Technology and Safety,
School of Navigation, Wuhan
University of Technology
Wuhan, China
LinyingChen@whut.edu.cn

Linhao Xue
School of Navigation, Wuhan
University of Technology
Wuhan, China
xue_lh@whut.edu.cn

Yangying He
School of Intelligent Sports
Engineering, Wuhan Sports
University
Wuhan, China
yangyinghe@whsu.edu.cn

Pengfei Chen
State Key Laboratory of Maritime
Technology and Safety,
School of Navigation, Wuhan
University of Technology
Wuhan, China
Chenpf@whut.edu.cn

Junmin Mou
State Key Laboratory of Maritime
Technology and Safety,
School of Navigation, Wuhan
University of Technology
Wuhan, China
Moujm@whut.edu.cn

Yamin Huang
State Key Laboratory of Maritime
Technology and Safety,
School of Navigation, Wuhan
University of Technology
Wuhan, China
YaminHuang@whut.edu.cn

*Abstract*—**Sailing in formation has the benefits of drag reduction. In current studies of hydrodynamic analysis of ship formations, the impacts of speed and spacing between adjacent ships on total resistance are seldom considered. To estimate the weight of different factors in formation on total resistance variation, the impacts of speed, longitudinal distance, and transverse locations on the observed total resistance of formations are investigated by analyzing hydrodynamic data in tandem, parallel, and triangle formation. The relation between resistance variation and speed is revealed. The regression analysis results on different formations indicate the differences between longitudinal spacing and transverse impacts. The regression formulation can be adopted to predict total resistance in formations.**

*Keywords—drag reduction, formation, regression analysis*

## I. INTRODUCTION

Nowadays, saving energy, reducing atmospheric pollutant emissions, and lowering carbon emissions are key concerns in the shipping industry. Increasingly, scholars are focusing on reducing ship resistance to save energy. Inspired by observing and analyzing duck flock swimming behavior [1], scholars have drawn insights from biomimicry and begun researching drag reduction through ship formations.

Chen et al. [2] studied the wave interference characteristics of two ships sailing in parallel and following each other and a three-ship "V" formation in shallow water using the bare hull of Series 60. The results indicate that when the two ships follow each other, the wave resistance for both ships decreases. In a three-ship "V" formation, the waves from the trailing ship provide additional thrust, significantly reducing the wave resistance of the leading ship. However, the additional reactive force from the wave crests of the leading ship increases the resistance of the trailing ship. Zheng et al. [3] used the second-order source method based on the Dawson method to calculate the wave resistance of four Wigley ships in three common formations: single-ship, two-ship formation, and three-body ship formation. They identified optimal ship formations for drag reduction in different speed ranges, and adjusting the relative positions of the ships in the Wigley formation can achieve drag reduction. Qin Yan et al. [4] first performed a numerical analysis of the drag characteristics of a single Wigley ship at different speeds. They compared the results with the hydrodynamic performance of a "train" formation at various longitudinal spacings. The analysis showed that, under all conditions, the total drag of the train formation was about 10% to 20% less than that of a single ship. For lower speeds, reducing the longitudinal distance can achieve drag reduction, but at higher speeds, increasing the longitudinal spacing helps maintain drag reduction. Liu et al. [5] used CFD to study the drag reduction effects of a KCS ship model in a twin-ship "train" formation at different speeds, showing that the drag reduction for the following ship could reach up to 24.3%. He et al. [6][7] focused on the hydrodynamic performance of three-ship formations at low speeds, analyzing linear, parallel, and triangular formations with equal and unequal spacing. The optimal ship formation configuration for drag reduction under different formations was ultimately identified. A regression model [8] was also developed to predict total resistance in different formation systems. Meanwhile, machine learning methods have also been applied to vehicle platooning problems to predict the drag of each vehicle in platoons of varying numbers (varies from 2 - 4). In summary, sailing in formation has the potential for drag reduction. Existing work [9][10][11] mainly focuses on observing drag reduction benefits at different speeds and formations configurations. However, the impact of factors on the resistance reduction of ship formation is unclear. Further research should be investigated to understand how different factors affect the total drag in ship formations.

Therefore, this paper aims to clarify the direct relationship between speed, spacing, and total resistance in ship formations. The primary innovation of this paper lies in employing regression analysis to quantitatively assess the ship formation CFD database, aiming to determine the extent to which speed and distance influence the resistance encountered during ship formation navigation.

The main contributions of the paper are as follows:

National Natural Science Foundation of China

- Quantitative analysis and estimation of the effects of factors (speed, longitudinal distances, and transverse locations) on total resistance in formations are provided.

- A regression model is established to predict the total resistance of the multi-ship formation system.

Subsequently, the datasets investigated in our research are introduced in Section II. Section III explains the proposed research approach. The analysis results for the impacts of different factors are presented, and the regression model is built in Section IV. In the last, Section V concludes the main findings and recommendations for further research.

## II. DATA DESCRIPTION

### A. Source of data

In this research, the dataset consists entirely of CFD simulation data. All the simulation is calculated via commercial software STAR CCM+ V13.06. Before the systematic simulation, verification and validation have been done. Therefore, the accuracy of the CFD results is guaranteed.

### B. Studied ship in dataset

In our CFD simulation conditions, the three-ship isomorphic formation is composed of three identically bare hulls of the full-swing tugboat 'WillLead I'. The parameters of the ship are shown in Table 1, and the side view is presented in Figure 1.

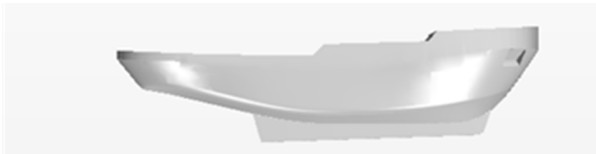

Fig. 1. Side view of the bare hull of 'Willlead I'

TABLE I. PARAMETERS OF 'WILL LEAD Ⅰ'

|  | λ | $L_{OA}$(m) | $L_{PP}$(m) | B(m) | T(m) | $A_S$(m²) |
|---|---|---|---|---|---|---|
| **Full scale** | 1.00 | 34.95 | 30.00 | 10.50 | 4.00 | 432.41 |
| **Model scale** | 17.475 | 2 | 1.72 | 0.674 | 0.211 | 0.672 |

### C. Data composition

The dataset comprises CFD simulation results in four different formation configurations: tandem formation, parallel formation, right triangle formation, and general formation. Besides, the longitudinal distance ($ST_1$, $ST_2$) and transverse locations ($SP_1$, $SP_2$) are different. The illustration of formation configurations is shown in Figure 2. The range of $ST_1$, $ST_2$, $SP_1$, $SP_2$ is shown in Table 2. In tandem formation, $SP_1$ equals $SP_2$ as zero; in parallel formation, $ST_1$ equals $ST_2$ as zero. In a right triangle formation, the bow of ship 2 aligns with ship 3, and the centerline of ship 1 aligns with ship 2. In a general triangle formation, the bow of ship 1 aligns with ship 3.

TABLE II. RANGE OF $ST_1$, $ST_2$, $SP_1$, $SP_2$

| Configuration | $ST_1$(m) | $ST_2$(m) | $SP_1$(m) | $SP_2$(m) |
|---|---|---|---|---|
| **Tandem** | 0.25-2.0 | 0.25-2.0 | / | / |
| **Parallel** | / | / | 0.1685-2.022 | 0.337-2.696 |
| **Right triangle** | 0.25-1.0 | 0.25-1.0 | 0.1685-0.674 | 0.1685-0.674 |
| **General triangle** | 0.25-1.0 | 0.25-1.0 | 0.1685 | 0.337-0.5055 |

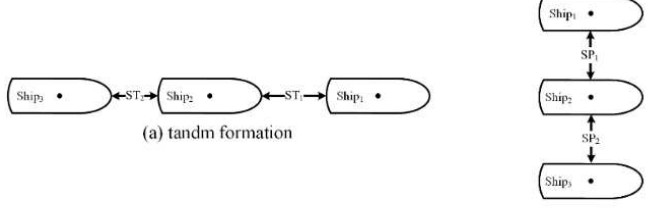

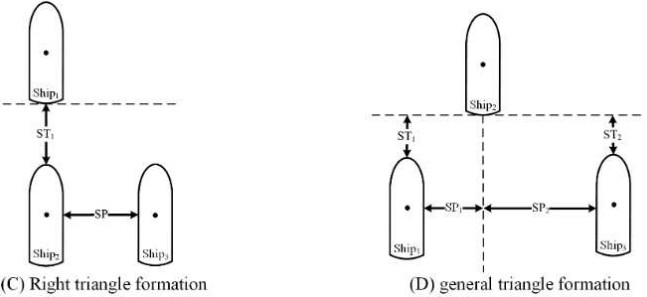

Fig. 2. Illustration of formation configurations

## III. METHODOLOGY

This research uses CFD data to investigate the influence of speed and spacing between adjacent ships in formations. In this section, the no-dimension coefficients of the formation and speed are illustrated in the coordinate system. The data analysis method is introduced, including data preparation.

### A. Dimensionless coefficients and coordinate system

The coordinate system to describe the motion and resistance of the formation is presented in Figure 3. The space-fixed coordinate system $O_o$-$X_oY_o$ and the ship-fixed coordinate system O-xy constitute the global coordinate system. The space-fixed coordinate system is used to describe the motion of the formation, and the ship-fixed coordinate system is used to describe the resistance of the ship in formation. In the space-fixed coordinate system, the Xo direction points to the true north. In the ship-fixed coordinate system, the x direction indicates the bow of ship, and the y direction points to the starboard side. The direction of no-dimension coefficients of resistance, including drag and the lateral force, are provided in Figure 3. $X^{'}$ is the no-dimension coefficient of longitudinal resistance, and the direction of $X^{'}$ from the bow to the stern is opposite to the x direction. $Y^{'}$ is no dimension coefficient of lateral force and the direction of $Y^{'}$ from the portside to the starboard side agrees with the y direction. The total dimensionless longitudinal resistance coefficient $X^{'}_{total}$ can be obtained by summing $X^{'}$ of each ship in the formation. In a similar vein the total dimensionless longitudinal resistance coefficient $Y'_{total}$ can be obtained by summing $Y'$ of each ship in the formation system. The equations of $X^{'}_{total}$ and $Y^{'}_{total}$ as follows:

$$X^{'}_{total} = \sum_{i=1}^{3} X^{'} \qquad (1)$$

$$Y^{'}_{total} = \sum_{i=1}^{3} Y^{'} \qquad (2)$$

In the research, the fleet is assumed to sail in calm water. Therefore, the impact of wind and current is not considered.

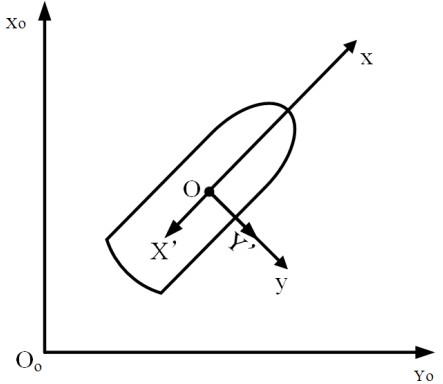

Fig. 3.   Illustration of the coordinate system

## B. Data preparation

Since the CFD simulation via STAR CCM+ V13.06 needs to set up the numerical and physical layouts, longitudinal distances ($ST_1$, $ST_2$) and transverse locations ($SP_1$, $SP_2$) mentioned in section 2 could only represent the geometric relationship between neighbor ships. To facilitate the learning of the characteristics of the data during the regression analysis, the longitudinal and transverse locations in the dataset are rearranged. $ST_i$ is specified to be the sum-of-signs value, when ship $i$ is in front of ship $i+1$, and $ST_i$ is specified to be the opposite of the geometric value when it is behind ship$_{i+1}$, $SP_i$ is specified to be the sum-of-signs value of geometric value when ship $i$ is located on ship$_{i+1}$'s port side, and $SP_i$ is specified to be the opposite of the geometric value when ship$_i$ is located on ship$_{i+1}$'s starboard side.

## C. Data analysis method

Figure 4 presents the steps of the regression analysis method.

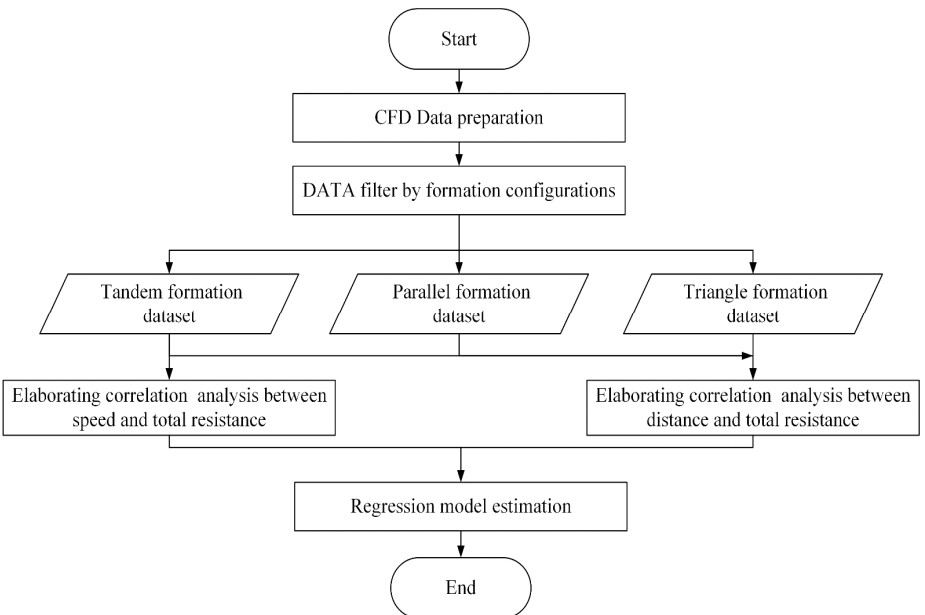

Fig. 4.   Flow diagram of regression analysis.

The hydrodynamic dataset of the ship formation is divided into different subsets to analyze the effects of speed and spacing between ships. The impacts of both longitudinal distances and lateral locations are considered on the total resistance of the ship formation system. The total resistance variations among the formation of different speeds have been observed. However, the direct relationship between total resistance and speed is still not revealed. The relationship between total resistance and speed is expected to be found using the tandem formation dataset. During the quantitative analysis of the speed impacts on total drag in tandem formation, the tandem formation dataset is split into subsets of different $ST_1$ distances. Then, a correlation analysis between total resistance and speed is performed to highlight the strength of the correlation and determine which speed criterion more effectively characterizes variations in total resistance.

Three steps are taken to quantify the impacts of longitudinal spacing and lateral locations. Firstly, the dataset is divided into six subsets based on different speeds. Each subset is further categorized into tandem formation, parallel formation, and triangle formation. After that, regression analysis is conducted on subsets of total resistance data at uniform speeds. The results will reveal if the impacts of ST and SP differ across various fleet speeds. Finally, overall functions will be defined to describe ST and SP impacts, incorporating speed variations, with coefficients estimated from the entire dataset.

After correlation analysis with different factors, a model for the formation system's total drag regression formulation is developed, including the five features: speed, $ST_1$, $ST_2$, $SP_1$, and $SP_2$. Multivariate polynomial and ridge regression methods are combined to build a regression model. Polynomial regression is a method of regression analysis based on polynomial functions for fitting non-linear relationships in data. Compared with linear regression, polynomial regression could model the non-linear characteristics of the data by introducing polynomial terms, thus increasing the flexibility and applicability of the model. In practice, data has many features, and polynomial regression for

a single feature performs poorly on fitting data with many features. Thus, multivariate polynomial regression is used in this study to fit the total resistance dataset of ship formations.

In practical applications of using multivariate polynomials for regression analysis, choosing the polynomial degree carefully is crucial. If the degree is too low, it may result in poor fitting performance. On the other hand, if the degree is too high, it can lead to overfitting issues where the model fits noise in the data rather than capturing the underlying trends. Therefore, when employing multivariate polynomials for regression analysis, it's crucial to select the degree of the polynomial judiciously. To address potential overfitting issues and improve the accuracy of data fitting when using multivariate polynomials to establish regression equations, this study introduces a combined approach of ridge regression with multivariate polynomial regression to build the regression model. Ridge regression is an improved least squares estimation method that addresses multicollinearity by introducing an L2 norm penalty term, thereby enhancing model stability and generalization capability. The penalty term is λ times the sum of the squares of all regression coefficients (where λ is the penalty coefficient). Combining ridge regression with multivariate polynomial regression can effectively control the complexity of the model and reduce the risk of overfitting by introducing a penalty term. This is particularly beneficial when input features are highly correlated or when the condition number of the data matrix is high. Such stability helps mitigate numerical computation issues that may arise in multivariate polynomial regression, thereby enhancing the reliability of the model.

## IV. RESULTS AND DISCUSSION

In this section, the impacts of speed, longitudinal location, and transverse spacing are analyzed to estimate the final regression model.

### A. Variation of drag due to speed

To estimate the relationship between speed and total resistance, the total resistance of the formation and the speed is provided in Figures 5 to 8. In these plots, the relationship between speed and total resistance of tandem formation under different longitudinal spacing $ST_1$ and $ST_2$ is depicted. Simultaneously, the combined resistance experienced by three individual ships sailing alone at various speeds is also provided.

The blue dots in the graph represent the total resistance experienced by the formation system, while the red line indicates the combined resistance experienced by three individual ships sailing alone at different speeds. The purpose of marking the red line on the graph is to determine whether a three-ship tandem formation can achieve a resistance gain compared to three ships sailing individually. When $ST_1$ is set as 0.25 $L_{OA}$, and 2.0 $L_{OA}$ both ships sailing individually and ships sailing in formation, the resistance of 'WillLead Ⅰ' ships decreases as ship speed increases. Simultaneously, the formation system benefits from resistance gains, with the maximum gain occurring at a speed of 0.212 m/s, reaching up to 4.85% in maximum resistance reduction.

When $ST_1$ is set as 0.5 $L_{OA}$, the total resistance observed during sailing in formation decreases as speed increases. However, the formation system did not gain resistance benefits.

Instead, it experienced resistance amplification, with the maximum increase reaching 119.3% at $ST_1 = 0.5$ $L_{OA}$.

When $ST_1$ is set to 1.0 $L_{OA}$ and 1.5 $L_{OA}$, the formation system experiences resistance gains. However, as ship speed increases, the resistance benefits gradually decrease. Additionally, when $ST_2$ is smaller than $ST_1$, the resistance benefits of the formation system nearly disappear as the ship speed increases to 0.424 m/s.

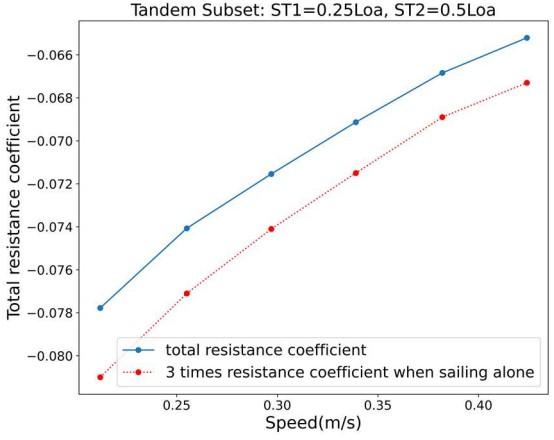

(a) $ST_2 = 0.5 L_{OA}$

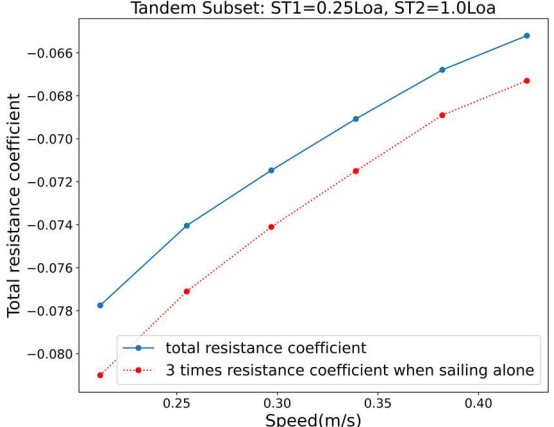

(b) $ST_2 = 1.0 L_{OA}$

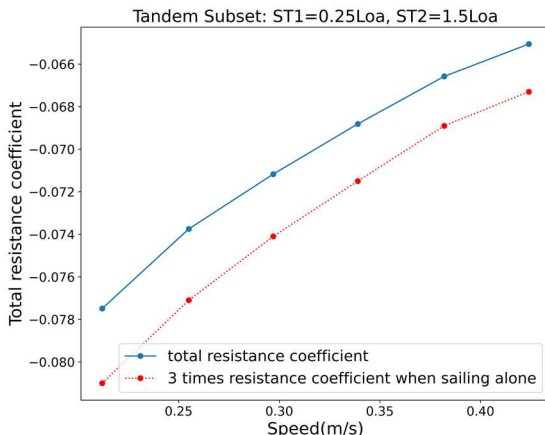

(c) $ST_2 = 1.5 L_{OA}$

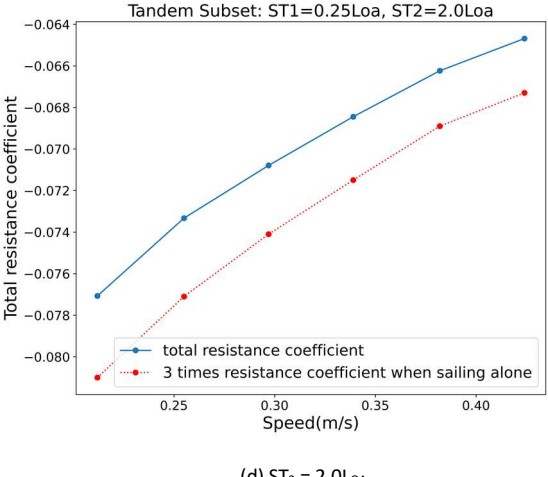

(d) $ST_2 = 2.0L_{OA}$

Fig. 5. Variation of resistance coefficient with speed when $ST_1 = 0.25 L_{OA}$

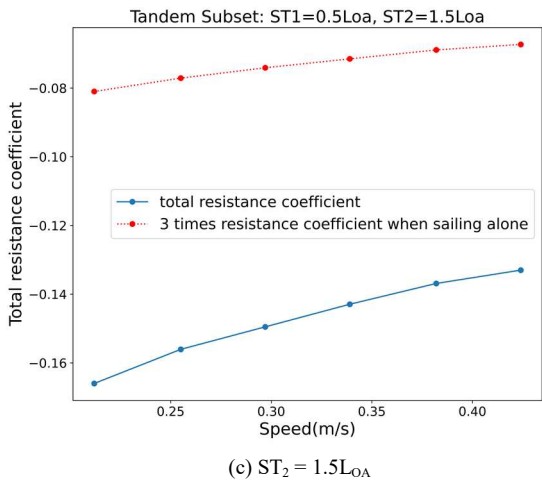

(c) $ST_2 = 1.5L_{OA}$

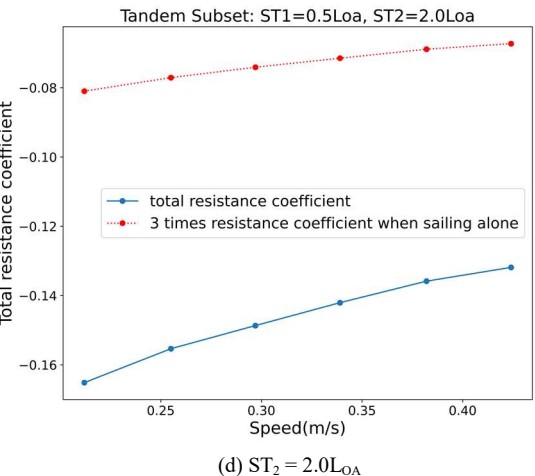

(d) $ST_2 = 2.0L_{OA}$

Fig. 6. Variation of resistance coefficient with speed when $ST_1 = 0.5 L_{OA}$

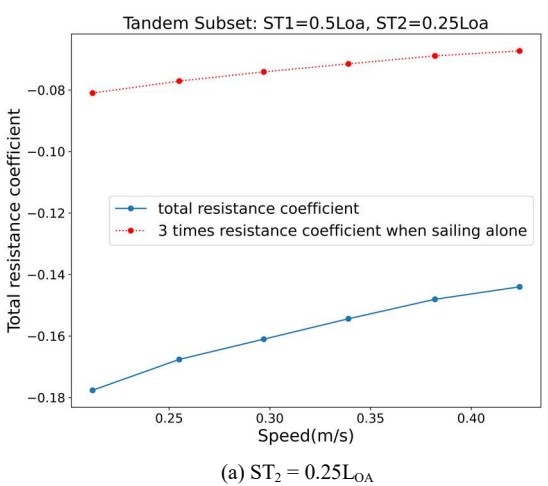

(a) $ST_2 = 0.25L_{OA}$

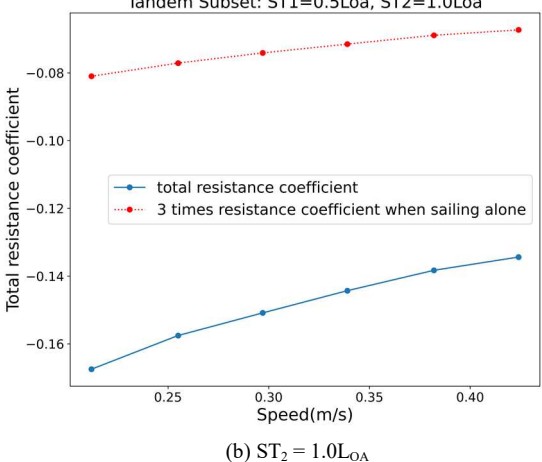

(b) $ST_2 = 1.0L_{OA}$

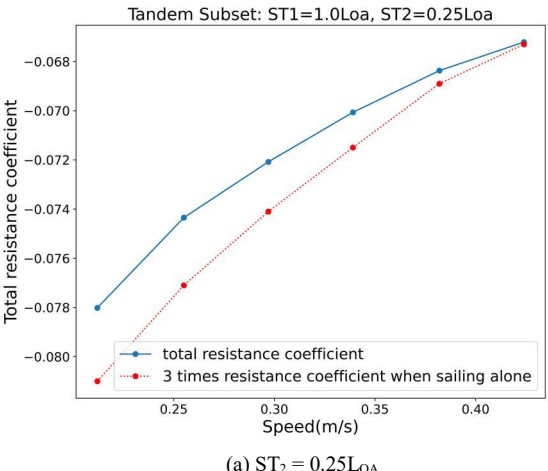

(a) $ST_2 = 0.25L_{OA}$

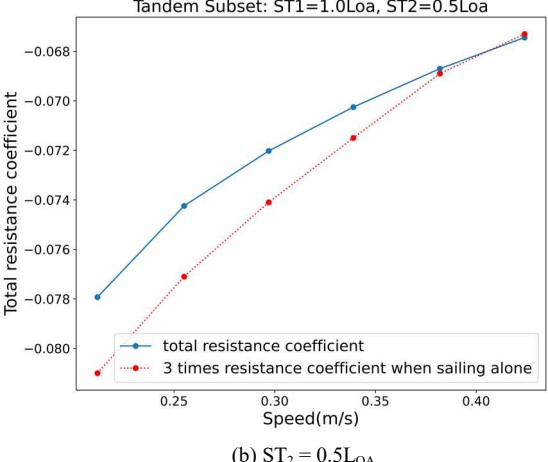

(b) $ST_2 = 0.5L_{OA}$

TABLE III. CORRELATION ANALYSIS BETWEEN RESISTANCE AND SPEED

| $ST_1$ | $0.25L_{OA}$ | $0.5\ L_{OA}$ | $1.0\ L_{OA}$ | $1.5\ L_{OA}$ | $2.0\ L_{OA}$ |
|---|---|---|---|---|---|
| $C_U$ | 0.99 | 0.91 | 0.98 | 0.98 | 0.99 |

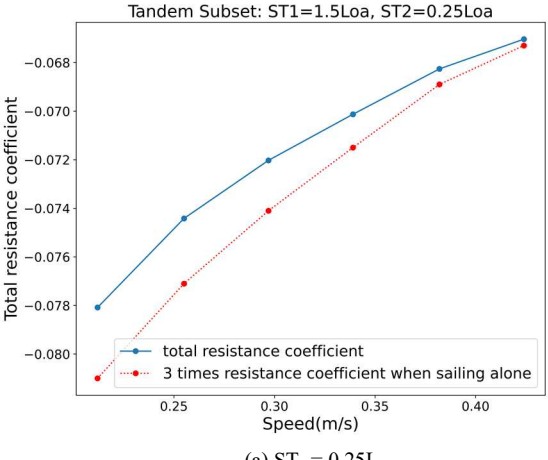

(a) $ST_2 = 0.25L_{OA}$

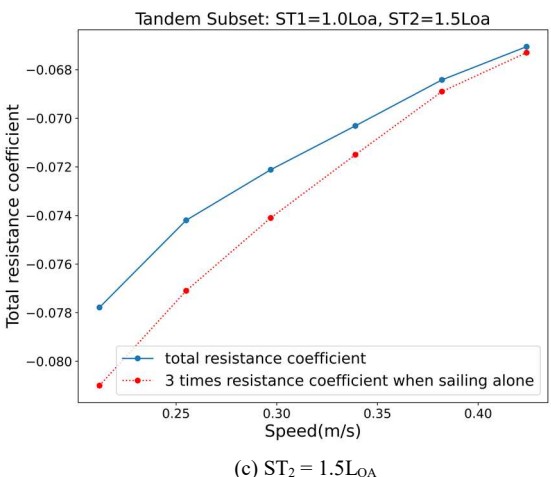

(c) $ST_2 = 1.5L_{OA}$

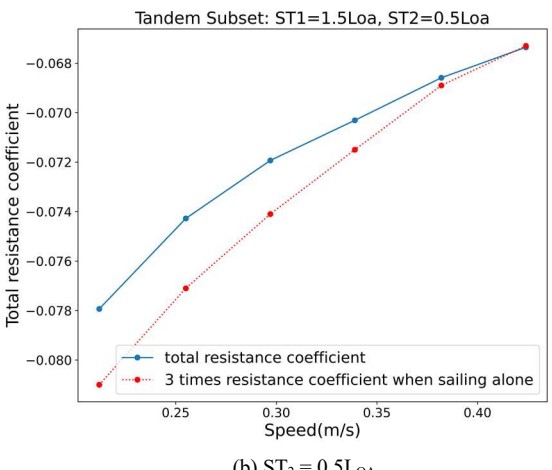

(b) $ST_2 = 0.5L_{OA}$

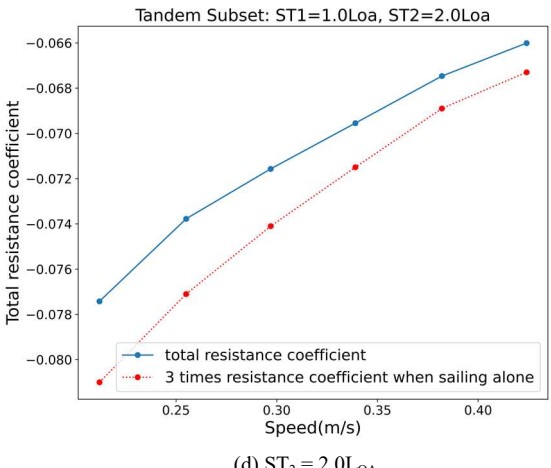

(d) $ST_2 = 2.0L_{OA}$

Fig. 7.  Variation of resistance coefficient with speed when $ST_1 = 1.0\ L_{OA}$

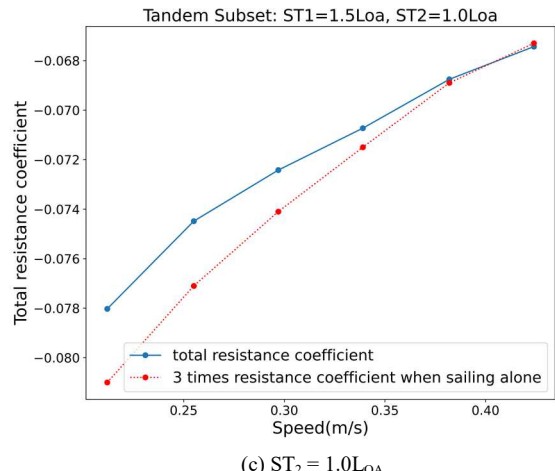

(c) $ST_2 = 1.0L_{OA}$

In tandem formations, the transverse distances SP1 and SP2 and the lateral forces do not affect the total resistance of the formation system. A correlation analysis between total resistance and speed of the formation is conducted. The results are shown in Table 3. All correlation coefficients are significant at 0.01 level of p-value(two-tailed).

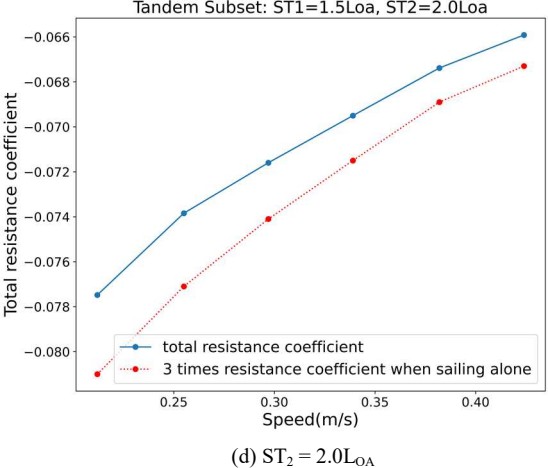

(d) $ST_2 = 2.0L_{OA}$

Fig. 8. Variation of resistance coefficient with speed when $ST_1 = 1.5\ L_{OA}$

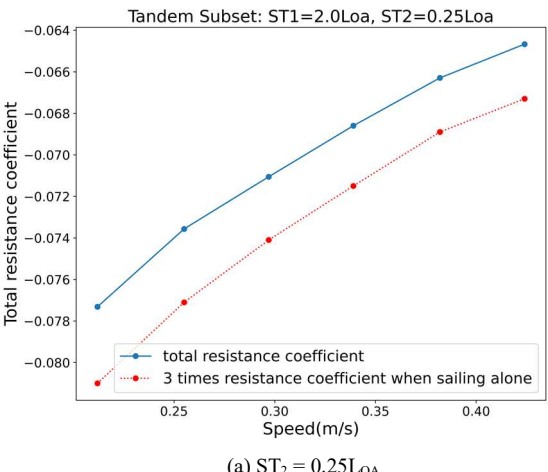

(a) $ST_2 = 0.25L_{OA}$

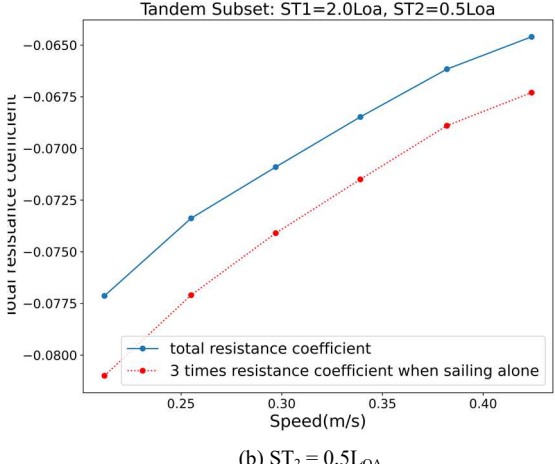

(b) $ST_2 = 0.5L_{OA}$

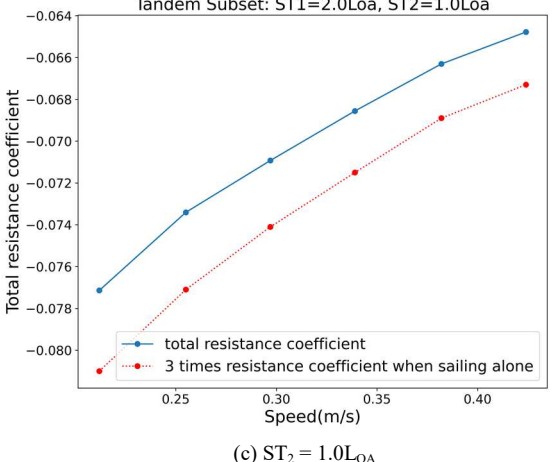

(c) $ST_2 = 1.0L_{OA}$

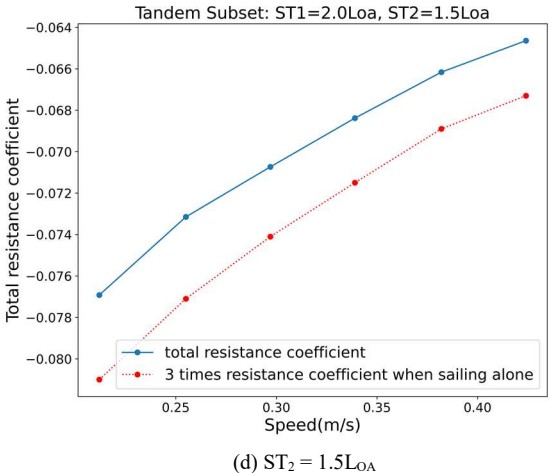

(d) $ST_2 = 1.5L_{OA}$

Fig. 9. Variation of resistance coefficient with speed when $ST_1 = 2.0\ L_{OA}$

### B. Quantification of longitudinal spacing and transverse location

This section presents regression analysis results of spacing in adjacent ships in formations. The results reveal the impact of spacing in adjacent ships ($ST_1\ ST_2\ SP_1$, $SP_2$) on total resistance. In tandem formation, the transverse locations $SP_1$, and $SP_2$, are set as zero. Besides, both $ST_1$ and $ST_2$ are varied from $0.25L_{OA}$ to $2.0\ L_{OA}$. So, there is no need to standardize the coefficients of $ST_1$ and $ST_2$ when calculating the coefficient in tandem formation subset.

Similarly, $ST_1$, and $ST_2$, are set as zero in parallel formation. The effect of standardizing the coefficients of $ST_1$ and $ST_2$ before calculating the coefficient in the parallel formation subset is insignificant. However, longitudinal distance and transverse spacing existed between the neighboring ships in the triangle formation. The longitudinal distance is much bigger than the transverse spacing. The unstandardized coefficients can not be compared directly. However, the standardized coefficients, derived from standardized regression analysis, are adjusted so that the variances of the variables are 1. in triangle formation. Thus, considering the need for standardizing correlation analysis under triangular formation configurations, standardized

regression analysis is adopted for correlation analysis in all conditions to unify the correlation coefficient analysis operations.

The whole data set of the total resistance of tandem formation is split into different subsets with the same speed. The coefficients of $ST_1$ and $ST_2$ for the total drag variable in each subset are presented in Fig 10. The results clarify whether $ST_1$ or $ST_2$ significantly impact total resistance in this multivariant regression model.

Two comparisons are made to interpret the estimated standardized coefficients. For tandem formation within the same subset, the weights of $ST_1$ and $ST_2$ are compared. The impact of $ST_1$ on total resistance is more significant than that of $ST_2$.

The other comparison involves analyzing coefficients for different speed groups, which reveals how external impacts vary among ships at different speeds. This analysis shows distinct trends in the effects of $ST_1$ and $ST_2$. on total resistance is flat when the speed gets bigger. The correlation coefficient of $ST_2$ ranges between -0.083 and -0.075, indicating a negative correlation between $ST_2$ and total resistance in tandem formation. With $ST_2$ increasing, total resistance tends to decrease. It is suggested that increasing $ST_2$ can help the formation system reduce total resistance. However, the influence of $ST_2$ on total resistance is instinctive. The correlation coefficient of $ST_1$ ranges between 0.42 and 0.435, indicating a positive correlation between $ST_1$ and total resistance in tandem formation. With $ST_1$ increasing, the formation system may gain energy benefits. It is suggested that decreasing $ST_1$ can help the formation system reduce total resistance. However, the influence of $ST_1$ on total resistance is significant. Thus, choosing $ST_1$ carefully is more effective than selecting $ST_2$ in obtaining total resistance benefits in tandem formation.

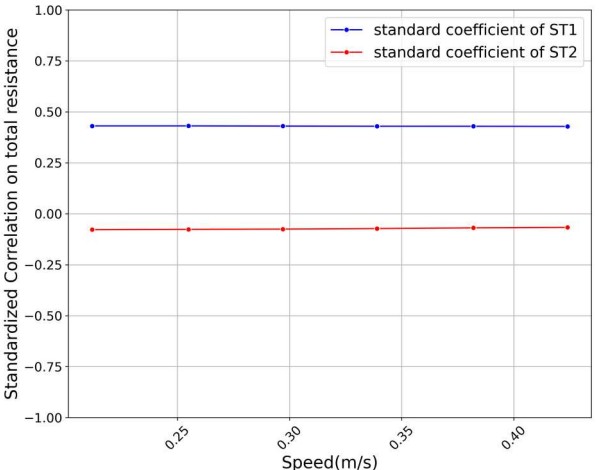

Fig. 10. The standardized coefficients of $ST_1$ and $ST_2$ on total resistance in tandem formation.

The whole data set of the total resistance of parallel formation is split into different subsets with the same speed. The coefficients of $SP_1$ and $SP_2$ for total resistance in each subset are presented in Fig 11.

Examining the standardized coefficients for parallel formation within the same subset allows for comparing the

effects of SP1 and SP2. For parallel formation, both SP1 and SP2 have a significant impact on total resistance. The impact of SP1 is slightly higher than that of SP2. In parallel formation, controlling the lateral spacing $SP_1$ between $Ship_1$ and $Ship_2$ is more effective in gaining resistance benefits compared to controlling the lateral spacing $SP_2$ between $Ship_2$ and Ship3. It also can be observed that the trends of both impacts of $SP_1$ and $SP_2$ on total resistance are undulatory with speed varying. The correlation coefficient of $SP_1$ ranges between 0.823 and 0.844, indicating a positive correlation between $SP_1$ and total resistance in parallel formation. With $SP_1$ increasing, resistance benefits tend to decrease. The influence of $ST_2$ on total resistance is positive. The correlation coefficient of $SP_2$ varies from 0.700 to 0.722, indicating a positive correlation between $ST_1$ and total resistance in tandem formation. With $ST_1$ increasing, the formation may gain resistance reduction benefits too.

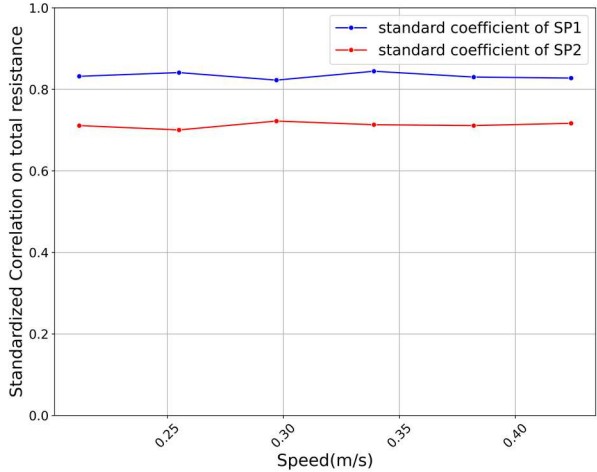

Fig. 11. The standardized coefficients of $SP_1$ and $SP_2$ on total resistance in parallel formation.

The whole data set of the total resistance of right triangle formation is split into different subsets with the same speed. The coefficients of ST and SP for total resistance in each subset are presented in Fig 12. Analyzing the standardized coefficients for right triangle formation within the same subset reveals that the impact of ST is less significant compared to SP Besides, the impact of both ST and SP on total resistance is positive. The Impacts of SP is more significant than ST. It also can be observed that the effect of ST on total resistance changes more gradually with speed compared to the impact of SP on total resistance. The correlation coefficient of ST ranges remains at 0.43, nearly unchanged, and the correlation coefficient of SP varies from 0.70 to 0.72, similar to the standardized correlation coefficient of $SP_2$ in parallel formation.

Regression models have been developed to quantitatively assess the effects of speed, ST, and SP on total resistance for tandem, parallel, and triangle formations. This paper presents the final regression models established using the complete dataset. Multivariant polynomial and ridge regression methods are combined to build the regression model. Due to the limited sample size, k-fold cross-validation was employed to enhance the robustness of the regression model.

The 4th-order regression functions are listed as equation (3)

$$
\begin{aligned}
X_{\text{total}} =\ & 0.01SP_1^4 - 0.13SP_1^3SP_2 + 0.81SP_1^3ST_1 + 0.81SP_1^3ST_2 + 1.6SP_1^3 + 0.12SP_1^2SP_2^2 + 0.6SP_1^2SP_2ST_1 + 0.6SP_1^2SP_2ST_2 \\
& -0.01SP_1^2SP_2U + 0.98SP_1^2SP_2 + 2.22SP_1^2ST_1^2 - 0.12SP_1^2ST_1ST_2 + 0.03SP_1^2ST_1U + 0.26SP_1^2ST_1 - 0.19SP_1^2ST_2^2 \\
& +0.01SP_1^2ST_2U + 0.26SP_1^2ST_2 + 0.05SP_1^2U - 1.28SP_1^2 - 0.24SP_1SP_2^3 + 0.85SP_1SP_2^2 + 2.01SP_1SP_2ST_1^2 \\
& -0.52SP_1SP_2ST_1ST_2 + 0.02SP_1SP_2ST_1U + 0.45SP_1SP_2ST_1 - 0.59SP_1SP_2ST_2^2 + 0.45SP_1SP_2ST_2 + 0.04SP_1SP_2U \\
& -0.74SP_1SP_2 + 3.0SP_1ST_1^3 - 1.11SP_1ST_1^2ST_2 + 0.08SP_1ST_1^2U - 2.08SP_1ST_1^2 - 1.19SP_1ST_1ST_2^2 - 0.06SP_1ST_1ST_2U \\
& +0.98SP_1ST_1ST_2 - 0.02SP_1ST_1U - 0.29SP_1ST_1 - 1.29SP_1ST_2^3 - 0.07SP_1ST_2^2U + 1.06SP_1ST_2^2 + 0.01SP_1ST_2U \\
& -0.29SP_1ST_2 - 0.02SP_1U - 0.45SP_1 + 0.1SP_2^4 + 0.27SP_2^3ST_1 + 0.27SP_2^3ST_2 + 0.02SP_2^3U + 0.03SP_2^3 + 2.41SP_2^2ST_1^2 \\
& -0.33SP_2^2ST_1ST_2 + 0.06SP_2^2ST_1U + 0.21SP_2^2ST_1 - 0.4SP_2^2ST_2^2 + 0.04SP_2^2ST_2U + 0.21SP_2^2ST_2 + 0.02SP_2^2U \\
& -0.35SP_2^2 + 3.26SP_2ST_1^3 - 1.18SP_2ST_1^2ST_2 + 0.23SP_2ST_1^2U - 2.6SP_2ST_1^2 - 1.27SP_2ST_1ST_2^2 + 0.09SP_2ST_1ST_2U \\
& +0.7SP_2ST_1ST_2 + 0.01SP_2ST_1U^2 + 0.04SP_2ST_1U - 0.06SP_2ST_1 - 1.38SP_2ST_2^3 + 0.08SP_2ST_2^2U + 0.8SP_2ST_2^2 \\
& +0.01SP_2ST_2U^2 + 0.07SP_2ST_2U - 0.06SP_2ST_2 + 0.02SP_2U^2 - 0.14SP_2U + 0.18SP_2 + 2.1ST_1^4 - 0.68ST_1^3ST_2 \\
& +0.12ST_1^3U - 4.17ST_1^3 - 0.75ST_1^2ST_2^2 - 0.02ST_1^2ST_2U + 1.18ST_1^2ST_2 - 0.08ST_1^2U + 2.5ST_1^2 - 0.76ST_1ST_2^3 \\
& -0.02ST_1ST_2^2U + 1.29ST_1ST_2^2 + 0.09ST_1ST_2U - 1.48ST_1ST_2 + 0.01ST_1U^2 + 0.01ST_1U - 0.17ST_1 - 0.83ST_2^4 \\
& -0.03ST_2^3U + 1.42ST_2^3 + 0.11ST_2^2U - 1.6ST_2^2 - 0.02ST_2U - 0.17ST_2 - 0.02U^4 + 0.01U^3 + 0.02U^2 + 0.15U + 0.62
\end{aligned}
\tag{3}
$$

The results of the estimation of the regression analysis are shown in Table 4. According to the regression analysis results, about 98.2% of the variance in the total power of the training systems can be explained by fleet speed. $ST_1$, $ST_2$, $SP_1$, $SP_2$ ($R^2$ is 0.982 for the whole dataset). Besides, speed has an estimate of 0.273, indicating a positive but relatively small effect on the dependent variable.

The standard error is 0.836, which is relatively large and suggests high uncertainty in the estimate. The t-statistic is 0.327, falling below common critical values (such as 1.96), indicating that the effect of this feature may not be significant. The standardized estimate of 0.327 aligns with the t-statistic, reinforcing that the standardized impact is also relatively modest. Feature $ST_1$ has an estimate of -0.171, reflecting a negative effect on the dependent variable. With a standard error of 0.157, the precision of this estimate is relatively high. However, the t-statistic of -1.089 is below common critical values, suggesting that the impact of $ST_1$ might also be non-significant. The standardized estimate of -1.089 confirms the direction of the effect but similarly indicates that its significance is weak. Feature $ST_2$ has an estimate of -0.167, suggesting a negative effect on the dependent variable. The standard error is 0.157, indicating high precision in the forecast. The t-statistic of -1.069 implies that this feature's impact may not be significant. The standardized estimate of -1.069 supports the direction of the effect but demonstrates that the impact is not substantial. Feature $SP_1$ is estimated at -0.501, indicating a strong negative impact on the dependent variable.

TABLE IV. ESTIMATION RESULTS OF THE FINAL REGRESSION MODEL

| | $R^2$ | F-state | Estimate | Std.error | t-stat |
|---|---|---|---|---|---|
| | 0.982 | 168.045 | 0.603 | 0.089 | 6.759 |
| $C_U$ | / | / | 0.273 | 0.836 | 0.327 |
| $C_{ST1}$ | / | / | -0.171 | 0.157 | -1.09 |
| $C_{ST2}$ | / | / | -0.167 | 0.157 | -1.07 |
| $C_{SP1}$ | / | / | -0.501 | 0.156 | -3.205 |
| $C_{SP2}$ | / | / | 0.128 | 0.159 | 0.806 |

The standard error is 0.156, which is relatively small, suggesting high accuracy in the estimate. The t-statistic of -3.205 exceeds common critical values, demonstrating that the effect of $SP_1$ is significant. The standardized estimate of -3.205 confirms that the impact remains strong even after standardization. Feature $SP_2$ has an estimate of 0.128, showing a positive but small effect on the dependent variable. The standard error is 0.159, which is relatively large, reflecting higher uncertainty in the estimate. The t-statistic of 0.806 is below common critical values, indicating that the effect of $SP_2$ is insignificant. The standardized estimate of 0.806 suggests that the impact is also small after standardization.

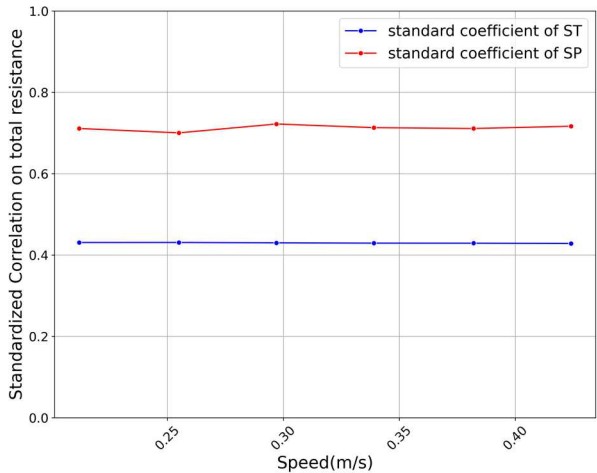

Fig. 12. The standardized coefficients of ST and SP on total resistance in triangle formation.

## V. CONCLUSION

The paper established a regression model to analyze the effects of factors including speed, longitudinal distances ($ST_1$, $ST_2$), and transverse locations ($SP_1$, $SP_2$) on the total resistance of ship formations derived from CFD data. The variation of total resistance in tandem formation due to speed can be observed.

The correlation analysis shows a strong correlation between speed and total resistance. The longitudinal spacing and transverse location impact on total resistance vary for different formation configurations. For tandem formation, both $ST_1$ and $ST_2$ have a more significant influence on total resistance. For parallel formation, the impact of both $SP_1$ and $SP_2$ slightly fluctuates with growing ship speed. However, for triangle formation, the impact of SP on total resistance shows a strong positive correlation. The ST impact on total resistance is negative. The regression analysis results revealed that about 98.2% of the variance in the total resistance of various ship formation systems was mainly explained by the factors that influenced its formation speed, $ST_1$, $ST_2$, $SP_1$, and $SP_2$.

This paper investigates the impact of different factors in the formation of total resistance. The estimated result indicates that more CFD data should be used in the regression analysis process. More intelligent methods can be used for regression analysis.

## ACKNOWLEDGMENT

The work presented in this study is financially supported by the National Natural Science Foundation of China under grants 52271364, 52101402, and 52271367.

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
