# OpenReview forum: "Impacts of speed and spacing on resistance in ship formations"
_IEEE.org/ICIST/2024/Conference — IEEE ICIST 2024 Conference Submission_

### Official Review · Reviewer_m1aS · 2024-08-21
**Impacts of speed and spacing on resistance in ship formations**

**Rating:** 7
**Confidence:** 4

**Review:**

This paper proposes a regression model to quantitatively analyze the impact of speed, longitudinal distances, and transverse locations on the total resistance of ship formations derived from CFD data. The research method used is correct. The research result has good practical significance. However, the legends in the simulation section are unclear, and the author should make appropriate adjustments to them. In addition, the research motivation should be clarified more clearly.

---

### Official Review · Reviewer_sqh7 · 2024-08-21
**Accept**

**Rating:** 7
**Confidence:** 5

**Review:**

This paper investigates the benefits of sailing in formation for drag reduction, which is a valuable topic with practical applications.By analyzing hydrodynamic data from tandem, parallel, and triangle formations, this study delves into the effects of speed, longitudinal distance, and transverse locations on the observed total resistance of formations. However, there are some potential art.eas for improvement.1.Many pictures in this article are not clear, please make adjustments.2.This article has some formatting issues. Please carefully checked.TABLE IV and Figure 1 have overlapping parts, please adjust the layout.

---

### Official Review · Reviewer_TrcS · 2024-08-25
**This work provides new insight into the development of regression models for analyzing the impact of speed and spacing on ship formation resistance**

**Rating:** 7
**Confidence:** 4

**Review:**

In the manuscript titled"Impacts of speed and spacing on resistance in ship formations"analyzes how speed and spacing affect ship formation resistance and provides a regression method for prediction.This work provides new insight into the development of regression models for analyzing the impact of speed and spacing on ship formation resistance.However, lack of  the contribution and innovation point is the major flaw of the study. The manuscript is well-organized and clearly stated. It is recommended to add sufficient background and literature comparison,leads to sufficient research motivation.Add a summary of innovative points and individual original work

---

### Decision · Program_Chairs · 2024-09-08

Accept (Oral)